# Breaking barriers for TB elimination: A novel community-led strategy revolutionizing tuberculosis case finding and treatment support in Senapati District Manipur-A quasi-experimental pre-post study protocol

Arangba S [1], Singh S[2], Nagarajan K[1], Malaisamy M[1], Watson B[1], Muanching L[3], Mattoo SK [4], Elangbam V[5], Singh WS[3], Ngade D[1], Ngaopuo A[1], Lungnalii KT[1], Serto T[3], Pfoze P[6], Nair D[1], Vignes Anand S[1], Elizabeth RK[7], Mark PS[8], Hanah RN[9], Yonuo P[8], Percy S[6], Padma Priyadarshini C[1], Kaur H [2]*

1 ICMR-National Institute for Research in Tuberculosis, Chennai, India, 2 Indian Council of Medical Research Headquarters, New Delhi, India, 3 National TB Elimination Programme, Manipur, India, 4 Central TB Division, New Delhi, India, 5 Regional Institute of Medical Science, Imphal, India, 6 Directorate of Health Services, Manipur, India, 7 National Health Mission, Manipur, India, 8 Senapati District Student Association, Manipur, India, 9 Senapati District Women Association, Manipur, India

* kaurh.hq@icmr.gov.in; harpreetkauricmr@gmail.com

## Abstract

### Introduction

Despite being the world's highest tuberculosis (TB) burden country, India still misses millions of TB cases annually. To address this issue, the India National Strategic Plan, following WHO strategy, promotes combining active case finding (ACF) with passive case finding (PCF) activities. National TB Elimination Programme (NTEP) began ACF campaigns thrice a year, targeting vulnerable populations. However, states like Manipur faced challenges in implementing and sustaining ACF activities due to resource constraints.

### Objective

To assess the impact of engaging student and women organizations (SAWOs) in improving TB case notifications, treatment adherence, and completion rate in NTEP, as well as to estimate the cost-effectiveness of the ACF intervention.

### Method

A quasi-experimental pre-post study is being conducted among individuals ≥15 years residing in Senapati District, Manipur, having two phases: preparatory and enhanced case finding and implementation of the ACF. Data is being collected and compared on TB case notification, treatment adherence, and outcomes beforeand after the

**Data availability statement:** No datasets were generated or analysed during the current study. All relevant data from this study will be made available upon study completion. Deidentified research data will be made publicly available when the study is completed and published.

**Funding:** This work is being supported by the Indian Council of Medical Research (ICMR) project funding grant number No. NER/82/2022-ECD-I, Proposal ID:2022-16180 to PI: Dr Stephen Arangba. The sponsors or funders had no role in the study design, data collection and analysis, decision to publish, or preparation of the manuscript.

**Competing interests:** The authors have declared that no competing interests exist.

intervention. Chi-square test will be used to test the statistical significance and logistic regression to identify the factors independently associated with the impact of intervention. Potential confounders at both patient and facility levels will be identified based on expert opinion and bivariate analysis. A multi-level logistic regression model will be used to control the confounding, with sensitivity analysis to ensure result robustness.Cost analysis will cover direct, indirect, medical, and non-medical costs for patients and health system. Incremental cost-effectiveness ratio per quality-adjusted life years gained will be evaluated.

## Discussion

This study introduces a novel community-led model involving SAWOsto improve TB case detection and treatment support, comprehensively addressing allfour pillars of 'END TB' strategy. The intervention is a community-based participatory research, emphasizing collaboration between researchers andcommunity to address TB control. The main activities of this intervention include community TB sensitization, ECF, ACF, treatment support and monitoring. This model could significantly impact TB control efforts, especially in resource-constrained settings like Manipur, offering valuable insights into ACF implementation and its economic implications.

## Introduction

Tuberculosis (TB) remains a significant global health challenge, causing substantial mortality and morbidity worldwide. In 2021, an estimated 10.6 million people were affected globally,resulting in 1.4 million deaths [1]. India alone accounted for 28% of the global TB burden, with 36% of TB-related deaths occurring among HIV-negative individuals [1]. Despite advances in TB diagnosis and treatment, a staggering 2.9 million TB cases were missing globally, including 17% from India, either undetected or unreported to National TB Elimination Program (NTEP) [2].

A significant factor contributing to the underreporting of TB casesin India is the overreliance onpassive case finding (PCF) strategies under the NTEP [3,4]. With PCF,the diagnosis of TB depends largelyon individuals seeking healthcare when they exhibit symptoms suggestive of TB [5]. According to the India National TB Prevalence Survey 2019–2021, 64% of TB-symptomatic individuals did not seek healthcare services [5] highlighting the urgent need for a comprehensive approach, that includes provider-initiated systematic screening for early diagnosis and treatment [6–8].

The WHO 'End TB Strategy' emphasizes systematic screening to identify the missing TB cases [9,10]. Aligned with this strategy,the Indian National Strategic Plan (NSP) 2017–2025 advocates for a combined approach of active case finding (ACF) alongside PCF to address the missing TB cases. Since 2017, the NTEPhas conductedACF activities in a campaign mode for two-week intervals thrice a yeartargeting vulnerable populations [11,12]. However, several states and districts,

including Manipur, encountered challenges in both implementing and sustaining these activities due to resource constraints [13–15]. The challenges in both executing and maintaining ACF initiatives underscore the need for innovative intervention strategies to effectively implement and sustain these efforts in India, a country burdened by high TB prevalence [16,17].

To address these challenges in implementing ACF, previous studies have often utilized Community Healthcare Workers (CHWs) and Accredited Social Health Activists (ASHA) workers, demonstrating their efficacy and cost-effectiveness [18–21]. Nevertheless, maintaining these efforts over time, particularly in India, has posed significant challenges [22,23]. CHWs/ASHA workers grapple with managing multiple health responsibilities, making it challenging to meet specific health program targets due to workload pressure [24]. Moreover, their uneven distribution across villages, coupled with language barriers and community acceptance issues, further complicates effective implementation [25,26]. Previous studies have also explored involving students in TB awareness and advocacy efforts, demonstrating successful engagement of elementary, secondary and National Service Scheme (NSS) students in case finding and treatment support [27–35]. However, to our knowledge, there remains a notable gap in engaging organized student and women organizations at the grassroots/village level for TB case finding and monitoring treatment adherence [36,37]. Student and women organizations are those organizations formed by individuals with defined roles and responsibilities to represent youth and women from their respective village, community, and district level to represent at a larger forum with similar goals and objectives to promote or celebrate a common interest. They work for the welfare of the society. They play a pivotal role in fostering community development, advocating for social justice, and addressing pressing societal issues. Their dynamic and multifaceted contributions stem from their ability to mobilize grassroots efforts, empower marginalized groups, and promote sustainable change. The organizations function as a youth and women wing in their respective villages, communities, and at the district level.

There is a wide spread of student and women organizations in India, especially in North-Eastern India, functioning with different names, with common goals and objectives. The student organization members include all the youths in their respective villages, irrespective of gender and profession, and the women organization members include all married women irrespective of professions. Both the organization leaders called executive staff were elected to the most influential person among their members in the village, community, and at the district level. Recognizing their potential, our study aims to involve SAWOs in Senapati district, Manipur to support NTEP in carrying out ACF activities and help in monitoring TB patients from the identification of symptoms to treatment completion. This study seeks to demonstrate the effective implementation and sustaining of ACF activities in the challenging geographical context of Manipur state. In addition, we will also assess the cost-effectiveness of this intervention to provide evidence for scalable community-based TB control strategies in resource-constrained settings.

## Methodology

### Study design

A quasi-experimental pre-post study is conducted across the Senapati District in Manipur to assess the impact of engaging SAWOs in improving TB case notifications, treatment adherence and completion rates as compared to existing routine strategies under the NTEP. Additionally, the study also aims to evaluate the cost-effectiveness of the SAWOs-led ACF interventions by estimating the cost per TB case diagnosed and treated through ACF.

### Study period

The actual study period is 18 months. However, the study period has been extended by another 6 months due to disruptions of activities in between the project period due to unexpected ethnic violence that erupted in Manipur, India on 3rd May 2023. The study was initiated on 27th March 2023 and is expected to be complete on 19th May 2025.

## Study implementers

The study is collaboratively conducted by the Indian Council of Medical Research-National Institute for Research in Tuberculosis (ICMR-NIRT), Chennai, ICMR Head Quarter,NewDelhi, Central TB Division (CTD), New Delhi,National TB Elimination Programme (NTEP), Manipur, National Health Mission (NHM) Senapati,Manipur,Senapati District Student Association (SDSA), Senapati District Women Association (SDWA), Regional Institute of Medical Sciences (RIMS) Imphal, Manipur and Directorate of Health Services (DHS), Senapati, Manipur.

## Study setting

Senapati district, situated in the northern part of Manipur, is characterizedby its hilly terrain. According to the 2011 census, the district comprises six sub-divisions, 686 villages, and a population of 479,148. The majority of the population (98.44%) resides in rural areas, with Scheduled Tribes (ST) accounting for 87.5% of the total [24]. There are no urban areas in the district except one census town. In 2016, the Government of Manipur divided the Senapati district into twodistricts, resulting in 7 sub-divisions [25,26]. As per the NTEP Senapati report, currently the district has 189 villages, 47,411 households, and a population of 285,404, with STs making up 92.74% of the population. It is the only district in Manipur where a Particularly Vulnerable Tribal Group (PVTG)resides. The district is predominantly inhabited byMaram, Mao, Poumai, Zelianrong, and Thangal tribes, each with distinct languages. Despite Manipuri being the official state language,many villagers do not speak or understand it. The villages are sparsely populated, located in mountainous regions, with limited all-weather road connectivity, posing logistical challenges, especially during the monsoonseason from June to September. According to the OKDISCD (Omeo Kumar Das Institute of Social Change and Development) household survey report, only 47% of the villages have all-weather road connectivity [38].

## Health infrastructure

As per the district NTEP report, there are one District TB centre(DTC), oneTuberculosis Unit (TUs),fourDesignated Microscopy Centres (DMCs) and elevenPrimary Health Institutions(PHIs). There are no private hospitals in the district. In Senapati district, there are one CBNAAT Machine, twoTruenatMachines, one FM microscope, and four ZN microscopes.

## TB burden status in the study district

The exact burden of TB in Manipur, particularly in the Senapati district, is yet to be precisely estimated due to the lack of prevalence studies. However, in a similar settingneighbouring district (Ukhrul district) in Manipurwhere over 70% of the population is tribal,reported a TB prevalence of 274/100,000, which increased to 358/100,000 after adjusting for Chest X-ray [39]. Another study using NFHS-4 data reported a prevalence of 710/100,000 in Manipur [40]. A modelling study also suggests a high TB infection rate in Manipur as compared to other states [41].

As per the NTEP report, TB notifications by PCF for the Senapati district were 162, 137, 145, and 190 cases in 2019, 2020, 2021, and 2022, respectively. However, the actual burden of TB in Manipur, especially in the Senapati district, is yet to be explored, and as no ACF activities have been conducted in the district or the state, as per the India TB report, thus these figures may not fully reflect the true burden [13,14].

## Exposure to TB risk factors in the study district

Various risk factors for TB are prevalent in Senapati District. The OKDISCD household survey reported that 35% of the Senapati district population lives below the poverty line, significantly higher than the state average of 22% for rural areas. Additionally, over 99% of households use solid fuels(wood, coal, kerosene oil, hay/leaves, and agricultural waste) for cooking which contributes to indoor air pollution and respiratory health risks [24]. According to NFHS-5, 46% of women and 64% of men aged ≥15 years in the Senapati district use tobacco in some form, further complicating TB control efforts

[42]. The district also faces an HIVprevalence of 0.45%, according to thedistrict-level estimates from the National AIDS Control Organisation (NACO), which increases susceptibility to TB infection and worsens treatment outcomes [43].

## Strategic approach

Given the highprevalence of TB,coupled with challenging terrain, inadequate health infrastructure, the infection rate in the state and the absence of ACF efforts at state and district levels, our strategy is to engage locally recruited SAWOs at the village level to enhance TB case detection and ensure treatment support until completion.

## Study population

Individuals≥15 years of age residing in the Senapati District of Manipur for at least a minimum of one month or more are included in our study. The exclusion criteria included mobile population/visitors, institutional populations, i.e., prisons, defense establishments, hospitals, nursing homes, hostels (except schools, colleges and offices),and areas where survey operations are considered not to be feasible like insecurity areas.

## Study phase and procedures

The study is conducted in two phases. The first phase ispreparatory and TB-Enhance Case Finding (TB-ECF)phase. The second phase is theimplementation of ACF, which serves as the intervention phase(Fig 1). The activities in eachphaseare detailed as follows:

## I. Preparatory and TB-ECF Phase

The preparatory and TB-ECFphase is currently ongoing. During this phase, the following seven activitiesare conducted: (1) building rapport and volunteers recruitment, (2) providing training to the volunteers, (3) assessing TB Knowledge, attitude, and practice (TB-KAP), (4) conducting a TB awareness campaign, (5) village mapping, (6) household enumeration and (7) performing TB-ECF.

## 1. Building rapport and volunteer recruitment

During this phase, the study teams explore the study areas and engage with SAWOs leaders at the district, tribal as well asvillage levels to establish rapport. The study team requests the SAWOs leaders to nominate one volunteer per 150 households from all the villages of the district. After obtaining a list of volunteers from SAWOs leaders, the study team screensthe volunteersfor eligibility, based on the specific criteria:(1) age 18 years and above,(2) member of localSAWOs, (3) fluent in the local languageand literate (able to read and write),(4) willing to undergo a 2-days training by the study team, (5) committed to staying in their area/village for at least one year,(6) able to provide written informed consent and (7) not currently diagnosed with active TB at the time of enrolment/consenting to become a volunteer. Eligible volunteers gave written consent, receiveddetailed study information,underwentscreening for TB symptoms, and had a chest X-ray to confirm their TB status before enrollment.

## 2. Training Volunteers

Eligible volunteers undergo 2 days of hands-on training. The training is provided by the TB research experts team (study team) including medical officers, public health experts, sociologists, social workers, statisticians, healtheconomists,and laboratory experts. All the training is conductedusing our specially designed training manual for this study. The training package includes (1) basic knowledge of TB (causes, symptoms, transmission, prevention, diagnosis, treatment aspects of TB, NTEP/DOTS strategy, risk factor, cough etiquette) (2) identification of presumptive TB cases (3) sputum collection and transportations (4) importance of monitoring during the treatment (5) importance of follow-up (6) basic counseling principles (7) data collection tools (8) thorough understanding of the study protocol.

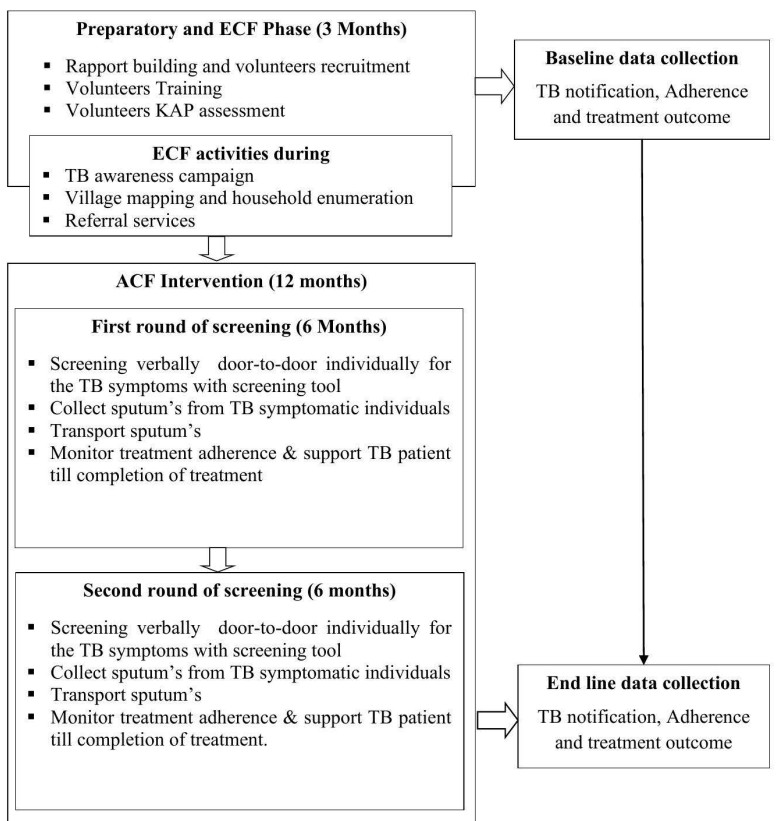

**Fig 1. Flow diagram of study phases and procedures.** The study is being conducted in two phases. (A) Preparatory and TB-Enhanced Case Finding (TB-ECF) phase, which included: rapport building and volunteer recruitment, volunteer training, TB knowledge, attitude, and practice (KAP) assessment, awareness campaigns, village mapping and household enumeration, and baseline data collection on TB notification, treatment adherence, and outcomes. (B) Active Case Finding (ACF) intervention phasecomprising two rounds of door-to-door verbal TB symptom screening, sputum sample collection from symptomatic individuals, sample transportation, patient follow-up, treatment adherence support, and endline data collection.

## 3. Asses TB-KAP

The study team (trainers) assesses the TB-KAP before and after completing the training to ensure that the volunteers have adequate knowledge about the TB before starting the screening process at the field level. Additional training is provided for another day to those volunteers who underperform on the TB-KAP test.

## 4. ConductTB awareness campaign

TB awareness campaignsare conducted to educate and sensitize the community about TB, reduce fear and stigma about TB, and also to inform the community that door-to-door ACF activities will take place. This exercise will help us gain more cooperation and trust from the community. Weplan to carry out forty TB awareness campaignsacross the district by merging/combining nearby villages after consulting with the village leaders and volunteers. The awareness campaign includes lectures in public gatherings with Information, Education, and Communication (IEC) activities. The study team, along with the volunteers and medical officers and their staff from all the community/primary health centers/institutions in the district, carry out the TB awareness campaigns.

## 5. Village mapping

Village mapping is done for all the villages in the district to establish distinct boundaries for each village and to establish designated ACF areas for volunteers. The trained volunteers carry out this activity to pinpoint the location of all house-holds within their designated area to ensure that no individuals or households are overlooked during screenings and to avoid repeated or duplicate screenings. Furthermore, this exercise helps the volunteers to efficiently navigate the house-holds and identify optimal access routes for planned visits.

## 6. Household enumeration

Household enumerationis carried out to estimate approximate baseline study populations for the upcoming interven-tions to ensure no individuals are overlooked during the rounds of screenings and also to avoid repeated or duplicate screenings. A trained volunteer visits door-to-door and linelists all the eligible participants and assignsUnique Individual Identification Numbers (UIIN) to each individual in their designated areas. TheUIIN is generated using REDCap software (electronic data capture tools). The software is installed on the volunteer's mobile device during the training session. The volunteers conduct individual interviewswith all the family members aged 15 years and above. For those individuals who are not there at the time of the first visit, two more attempts are made for the interview. After three attempts during the enumeration period and not being able to capture the data, another attempt is made during the first and second rounds of screening. The volunteerscollect socio-demographic details and information on common health problems faced by the individual in the last 5 years.

## 7. TB-ECF

We plan to do ECF during the preparatory phase to avoid delaying the diagnosis and treatment ofpeople with TB. We expect that some people who have TB symptoms may approach the research team/volunteers for consultation after receiving education and awareness about TB during the TB awareness campaign. For those who approach the research team/volunteers with TB symptoms during the preparatory phase, they areverbally screened for TB symptoms and referred to the nearest communityhealth center/primary healthinstitution/ primayhealthcentre/district TB center (CHC/PHI/ PHC/DTC) with a referral slip (Annexure-I in S1 File) for further diagnosis and treatment.

## II. ACF Intervention phase

This phase includes two rounds of screening. The total duration for both rounds of screening is twelve months, with each round lasting six months. The trained volunteers carry out door-to-door screening, individually assessing all the family membersaged >15 years for TB symptoms using specific screening tools tailored for each round. The first round screen-ing tool consists of information related tosocio-demographic details, TB symptoms, TB knowledge, TB history, the Indi-vidual'shabits oftaking tobacco and alcohol, and health-seeking behavior of people who have TB symptoms(Annexure-IV in S1 File). The second round screening tool consistsof information on TB symptoms and the Individual's habits of taking tobacco and alcohol (Annexure-V in S1 File). If any of the family members are not present during the first visit, two more attemptsare made at different times and days after consulting with the family members/neighbours who are present during the visit. Efforts are made to obtain the absentee's mobilecontact details and set an appointmentby phone based on their availability before attempting anothervisit. We aimto complete each round of active door-to-door TB symptom screening within the first month of starting the screening because the volunteers are present in almost all the villages, with each volunteer responsible for only 150 households. After completing each round of active door-to-door screening, volunteers stop visiting households for screening. However, unscheduled screening continuesas long as the volunteersareactive,and even after completing the project period, the active volunteers will continue to do the activities. Unscheduled screenings, we define as unplanned screenings where the community people (people who have TB symptoms) approach the volun-teers to be screened for TB symptoms anytime post-completion of the active door-to-door screening. Post-completion of

the project period, we proposed to continue the volunteer activities/service with the support of NTEP, NHM and DHS. The NTEP, NHM, and DHS will continue to monitor the activities of the volunteers.

In the first round, the volunteersscreened for the following TB symptomsusinground one screening tool which include TB symptoms like(1) Persistent cough for ≥2 weeks (2) Fever for ≥2 weeks (3) Loss of appetite (4) Night sweats for >2 weeks or unintentional weight loss (loss of 4.5 kg or >5 kg of the usual body weight over a period of 6–12 months) (5) Presence of blood in sputum any time during the last 6 months (6) Chest pain in last one month. When the volunteers identify any individual with TB symptoms, along with cough or only cough, two sputums, preferably mucopurulent,are collected in a sterile falcon tube, one on the spot and another the next day early morning. The volunteersdo not collect sputum from individuals without cough symptoms even if they have other TB symptoms,as they are unlikely to produce quality sputum. Such individuals are referred to the nearest health facilities for further investigation (Fig 2). Sputum is collected in an open well-ventilatedspace away from the crowded areas. Individuals with TB symptoms are given a sputum container with personal details written on its side (Name, age, gender, village name, type of sputum and volunteer name). The on-spot sputum is designated as 'S' and the next day's early morning sample as 'M'. All sputum is transported to the nearest PHIs/ PHCs/CHCs/DTC on the same day in a cold chain, following the NTEP guidelines (cool chain within 72 hours) by the trained volunteers. From the PHIs/ PHCs/CHCs, the project staff/ PHIs/PHCs/CHCs staff collect the samples on the same day and transport them to the DTC for investigations. All sputumsamples are tested usingCBNAAT. The individuals with TB symptoms whose two consecutive sputum (on-spot and early morning sputum) results are reported to be negative, but still have the TB symptoms are referred to the nearest chestX-ray centre for further investigations (Fig 2).

Once a confirmed TB case has been identified, the trained project staff informs/forwards the test result to the volunteer over the phone to inform the test result to the patient. The volunteers visit the patient's house to inform the patient of the test result from whom the sputum is collected. Also, with the consent of the patients, the volunteers inform the test result to the other family members who are residing in the same household. Subsequently, the volunteers encourage the family members to support the patient during their treatment and also encourage and counsel the Household contacts of pulmonary TB patients to go for further investigations and evaluationsat the nearest CHC/PHI/PHC/DTC to rule out active TB to start the TB Preventive Treatment(TPT) as per the NTEP guideline. All diagnosed TB patients are linked with the NTEP and treated as per the NTEP program.

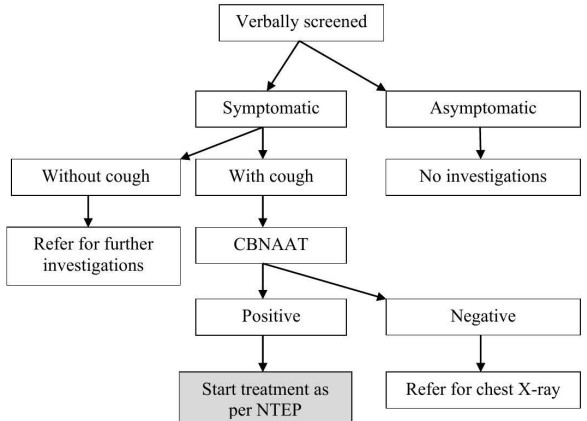

**Fig 2. TB screening and diagnostic algorithm.** Individuals aged >15 years were verbally screened by trained volunteers and categorized as symptomatic or asymptomatic. (A) Symptomatic individuals with or without cough: with cough underwent for CBNAAT testing. CBNAAT-positive cases were initiated on treatment as per NTEP guidelines. CBNAAT-negative individuals were referred for chest X-ray and without cough were referred for further investigation. (B) Asymptomatic individuals were not investigated further. Abbreviations: CBNAAT, Cartridge-Based Nucleic Acid Amplification Test; NTEP, National TB Elimination Programme.

All TB patients who are undergoing treatment, including the patient who has been identified through the routine program strategies,receivepsycho-social and adherence support from volunteers to complete their treatment. For non-adhering patients, volunteers help counsel them to complete the treatment by taking the prescribed medication on time. Any adverse events during treatment are reported by the volunteers to the district TB program officials. Throughout the journey of TB case identification and treatment, the volunteers act as liaisons between the health system and the community.

The second roundof screening is repeated after six months from the date of starting the first round of screening using the second-round screening tool. All the activities are repeated similar to the first round of screening.

### Comparator arm

Our comparator arm is the existing routine NTEP program of Senapati district, Manipur. The baseline comparison years of the study are 2019, 2020, 2021, and 2022, that is,before, during, and post-COVID-19 on the TB case notification rate, adherence rate, and treatment outcomes. As per the routine NTEP program data, the following Table 1is the rate of TB notification, adherence and treatment outcome for the year 2019–2022.

### Study population coverage and notification consideration

The year 2019 isconsidered as the reference period before the effect of COVID-19 on TB notification. In 2019, the annual target for TB notification was162,and 68% of the expected target was achieved. Similarly, 2021 is considered as the comparison period considering the effect of COVID-19. In 2021, the annual target for TB notification is145,and 76% of the expected target is achievedin the Senapati district. With this intervention, we aim to achieve at least80% of the target notification.

With the intervention, we aim to reach a TB notification rate of at least 130 per 100000 population. To detect this notification rate with a relative precision of 5% and alpha of 5%, the required sample size is about 39045. However, our study covers the entire population ≥15 years in the district.

**Table 1. 2019–2022 TB notification, adherence and treatment outcome.**

| Indicator | 2019 | 2020 | 2021 | 2022 |
|---|---|---|---|---|
| **Notification** | | | | |
| **Notification** (Achievementvs NTEP Target) | 162/240 (68%) | 137/190 (72%) | 145/190 (76%) | 190/155 (123%) |
| **Adherence** | | | | |
| **Adherence rate** | 76% | 87% | 99% | 95% |
| **Treatment outcome** | | | | |
| **Treatment Success rate** | 140 (147 cases in current facility) (95%) | 116 (123 cases in current facility) (94%) | 121 (132 cases in current facility) (91.6%) | 169 (173 cases in current facility) (97.6%) |
| **Loss-to-follow-up** | 0 | 1 (0.8%) | 1 (0.7%) | 1 (0.5%) |
| **Regimen changed** | 4 (2.7%) | 2 (1.6%) | 5 (3.7%) | 0 |
| **Death** | 3 (2%) | 3 (2.4%) | 4 (3.0%) | 1 (0.5%) |
| **Not evaluated** | 0 | 1(0.8%) | 1(0.7%) | 0 |
| **Wrongly diagnosed** | 0 | 0 | 0 | 1 (0.5%) |

**Source:** NTEP Senapati register & NIKSHAY portal.

Note: Cure and treatment completed are counted as treatment success.

Note: Cases in the current facility are the total number of TB patients initiated on treatment.

## Study recruitment/project status

Volunteers recruitment and training was started on 27ᵗʰ March 2023 and is ongoing to replace the drop-out volunteers. The first round of verbal TB symptoms screening at the community was started on 26ᵗʰ September 2023, and is ongoing, and the second round of screening was started on 1ˢᵗ April 2024, and is ongoing.

## Outcome

The study primary outcomeswill be the TB case notification rate, adherence rate, and treatment outcome. TB case notification rate is determined by the total number of TB cases notified in the district before and after the interventions. The treatment adherence rate is determined by the percentage of TB patients adhering to the treatment before and after the interventions. Treatment outcome is assessed bythe total number of patients cured, treatment completed, treatment failed, died, lost-to-follow-up, not evaluated, and treatment success before and after the interventions. All outcome indicators are measured according to the NTEP program definitions. Our study secondary outcomesinclude the direct and indirect costs per TB case notified, treatment success and quality adjusted life years (QALYs) gained.

## Data collection

**Tool used for data collection.** All the data is collected and managed using REDCap electronic data capture tools hosted at ICMR-NIRT. REDCap is a secure, web-based application designed to support data capture for research studies.

**TB KAP information.** TB KAP data is collected by the study team from the volunteers before and after the training, after obtaining written informed consent to become a volunteer. TB KAP data consist of socio-demographic details, TB knowledge, Attitude, and practice-related questions (Annexure-II in S1 File).

**Household enumeration information.** The household enumeration data is collected by trained volunteers from eligible community individuals. The data will consist of socio-demographic details and the common health problems faced by the individual in the last 5 years (Annexure-III in S1 File).

**First and second rounds of screening information.** First and second roundsof screening data are collected by trained volunteers from the eligible community individuals who have been enrolled duringthe household enumeration. The first round data consists of socio-demographic details, TB knowledge, TB history,Individual's habits of taking tobacco and alcohol, TB symptoms, and health-seeking behaviour of people who have TB symptoms (Annexure-IV in S1 File). The second round of intervention data consists of only TB symptoms and the Individual's habits of taking tobacco and alcohol (Annexure-V in S1 File).

**Laboratory sample and test result information.** Details of sputum samples received from the volunteers and sputum test results data are collected daily from the DTC lab technician/medical officer by the trained project staff. The project staff conveys the test result to the concernedvolunteers on the same day for further conveying it to the concerned studyparticipant.

**Baseline and endline information.** The baseline and endline data on TB case notification, adherence, and treatment outcomes are collected by the trained project staff during the preparatoryphase and end of the intervention from the DTC through the Nikshay portal and TB program registersbefore starting the intervention and after completing the intervention. Data discrepancies in the Nikshay portal and program registersarecompared and corrected accordingly along with the study team, NTEP program officials and WHO consultants.

## Cost information

Cost data is collected from both the health system and patient perspective. Trained field investigators conduct two rounds of data collection at participants' residences: first, during the intensive phase (initial two months of treatment) and again- upon completion of treatment. Data collection is conducted using a structured, pre-coded interview schedule. (Annexure -VI & VII in S1 File).

**Health system costs.** A health system perspective for economic evaluation considers only the recurring costs incurred by the health system, such as the cost ofprogram administration, human resources (staff salary), training, and implementation, honorarium for volunteers, monitoring, and supervision. In addition,information on laboratory consumables, stationeries, IEC documents, communication, review meetings, transport, and other costs is also collected from the intervention implementers and the existing TB program.

**Patient cost.** Patient cost is the out-of-pocket expenditure incurred by the individual who accesses treatment services for TB. This cost includesboth direct and indirect costs. Direct cost includes both medical and non-medical costs. Direct medical costs include doctors' consultation fees, money spent on investigations (X-ray, sputum examination) and drugs. Direct non-medical cost includes money spent on transportation and food. Indirect cost includesself-reportedworkabsenteeism and loss of wagesbecause of the TB illness for both the patient and their accompanying persons during treatment.

**Socio-economic status.** Information is also collected to measure the economic status of the participants through income, standard of living index (housing details and essentials) and wealth index (durables at home).

### Data quality assurance and confidentiality

To ensure data quality assurance, the study team ensures clarity of the study protocol and its procedures for all the project staff and volunteers during the training period, apart from the training package. A bi-monthly follow-up and monitoring visits are carried out by the study team along with the SAWO leaders to make sure that the work is done accurately and timely. During monitoring visits, we review consent forms, completeness of the data collection forms, and compliance with the study protocol. Any anomalies in the process are followed up by the field staff. The data is stored in secure servers at ICMR-NIRT. The REDCap system has built-in data validation and checks to ensure the consistency of the data. Also, regular quality control of the data is also done to ensure valid data collection.

### Data analysis plan

The baseline demographic and socio-economic characteristics of the study population (age, gender, occupation, education, income) will besummarized using means and standard deviations for continuous variables; frequencies and percentages for categorical variables. The household data, including the number of households surveyed and the total number of eligible individuals, will be summarized. The distribution of the common health problems reported by the participants will be obtained. The mean scores of TB knowledge, attitude, and practice from the pre-and post-training for volunteers will be calculated. The preand posttraining scores will be compared and statistical significancewill be tested using paired t-test.

TB case notification rates during the pre-intervention and post-intervention periods will be determined. Treatment outcomes (cure rates, treatment completion, loss-to-follow-up, regimen change, failure, mortality) pre and post intervention will be analyzed. These indicators pre and post intervention will be compared and chi-square tests will be used to test the statistical significance.

The individual level patient data will be obtained from the Nikshay portal. The potential confounders from the patient level and facility level data based on contextual factors such as facility size, patient volume, or baseline performance will be identified based on expert opinion and bivariate analysis. To control for confounding, we will include the confounders in the multi-level logistic regression model. Sensitivity analyses will also be conducted to assess the robustness of our findings. Logistic regression will be used to identifythe factors independently associated with the impact of the intervention.

Cost analysis, we will measure the total and average direct and indirect medical and non-medical costs incurred before (for diagnosis) and during treatment (for treatment) bythe patients. We will also measure the total and average cost incurred by the health system to treat a patient. The effectiveness is measured in terms of the number of TB cases detected and treated. Additionally, we will consider the incremental cost-effectiveness ratio (ICER) per QALY gained by

the intervention. The ICER will be calculated by dividing the difference in total costs between the intervention and comparator to determine whether the intervention is less-cost more-effective or more-cost less-effective.

## Ethical consideration

The institution ethics committee (ICMR-NIRT & RIMS) approved(No:234/NIRT-IEC/2022&No:A/206/REB/Prop(FP)186/114/25/2022) the study. The institution's ethics committee approves the participant information sheet and assent/consent form, which is used to obtain assent/consent from the study participants. Written or thumb impression consent/assent is obtainedonly from participants exhibiting TB symptoms during door-to-door screening. Oral consent/assent is obtained from other populations without TB symptoms. Participant information sheets are provided before obtaining consent/assent, and for those unable to read/understand, study volunteers will help in explaining the study details in the local language. Consent/assent is obtained after providing detailed information about the study.

For volunteers, written informed consent to become a volunteeris taken after meeting all the eligibility criteria to become a volunteer, before providing them with the training. An honorarium of ₹500 per month and a mobile recharge of ₹700 per quarter are given to the volunteers from the project for the service they render for the study. Apart from this, the volunteers are eligible to receive schemes provided by the NTEP program under the Government of India,including the informant incentive (₹500/- given on diagnosis of TB among referrals from the community to public sector health facility) and patient support incentive(₹1000 for drug-sensitive TB patients and ₹5000 for MDR-TB).

Personal Protective Equipment (PPE) Kits (face shield, glove, mask (N95), and hand sanitizer) were provided to the volunteers to be used during screening or household visits for counseling the TB patients. Also kits like vaccine carrier box, ice pack, falcon tube, parafilm M roll, tissue paper, sellotape, cotton, durable marker, and ziplock cover were provided to the volunteers for collecting and transporting the sputum samples. All the kits were handed over to the volunteers after completing arigorous two-dayhands-on training.

## Discussion

This study protocol introduces a novel community-led strategy aimed at enhancing TB case finding and treatment support. The primary objective of the study is to evaluatethe impact of engaging SAWOs inimproving TB case notifications, treatment adherence and completion rates as compared to existing routineNTEPstrategies. Additionally, the study aims to evaluate the cost-effectiveness of the SAWOs-led ACF interventionand address challenges encountered by the NTEP in India, particularly in the state of Manipur,regarding the implementation and sustainability of ACF activities. By engaging local SAWOs at the grassroot level, thestudy aims to enhance TB case detection efforts and ensure continuous treatment support until completion.

Aligned with India's National Strategic Plan, this project comprehensively addresses allfour pillars of TB control such as Detect, Treat, Prevent and Build. The study focuses on door-to-door screening to identifypresumptive TB cases and connect them to the health system for early diagnosis (Detect), early initiation of treatment for all the diagnosed TB patients (Treat), community education to dispel the misconception and stigma attached with the disease by conducting multiple TB awareness campaigns at the community level to break the chain of transmissions (Prevent), and community involvement in the decision-making process, involvement in program implementation, monitoring and evaluation (Build).

The outcomes of this study have the potential to significantly influence TB control efforts in India, especially in resource-constrained settings like Manipur. By involving SAWOs, the study introduces a novel approach that could be more sustainable and effective in reaching underserved populations and hard-to-reach areas because these organizations have deep-rooted connections within the community, which can help overcome barriers related to demography, language, trust and access to healthcare services. Our study addresses the social and behavioral factors that influence disease transmission as well as treatment outcomes by involving community members as active participants in TB control activities. This approach in the current study may not only improve TB case detection and treatment outcomes but also help

build local capacity for future health interventions. Thestudy also could provide valuable insights into the economic implications of implementing ACF strategies, which is essential for similar future planning and expanding TB control programs in high-burden, often resource-limited settings [44,45].

The engagement of SAWOs aligns with the community-based participatory research approach [46–48] emphasizes collaboration between researchers and community members to address TB control. Theprojectsupports the global effort to 'End TB' [34] by providing sustainable, evidence-based solutions for TB control in countries with similar settings. The study results will inform and facilitate the nationwide expansion of an effective intervention tailored to the Indian context and align with the principles of the NTEP to identify undiagnosed cases and control TB.

The key strength of this study is thus its comprehensive approach, which includes multiple components such as community sensitization, active case finding, treatment support and monitoring in addressing all aspects of TB control, thereby aiming to achieve a more holistic impact on TB prevention and care. However, several challenges and limitations should be considered. The study's success will rely on the active participation and commitment of the students and women's organization leaders and their members, which may vary depending on the local context and leadership. Additionally, the study's impact may be limited by the availability of resources and infrastructure in the study area.

Despite these challenges, this study represents an innovative modelfor TB control that could serve as a blueprint for other regions facing similar challenges. By leveraging existing community networks and resources, the study aims to improve TB case detection and treatment outcomes not only in Manipur but also potentially in other high-burden regions. Furthermore, continued research and evaluation will be essential to assess the long-term impact and scalability of this community-led approach.

## Dissemination plans

The importance of the study as well as the learnings/findings of the study will be disseminated to all the key stakeholders, including the NationalTB program, State & District health Officials, and through peer-reviewed journal publications.

## Supporting information

**S1 File.** Study tools and questionnaires used for data collection. Includes: **Annexure I** – Presumptive TB Case Referral Slip. **Annexure II** – TB Knowledge, Attitude, and Practice Questionnaire. **Annexure III** – Household Enumeration Questionnaire. **Annexure IV** – First Round TB Symptom Screening Tool. **Annexure V** – Second Round TB Symptom Screening Tool. **Annexure VI** – Part 1: Cost Identification Post-Treatment Initiation. **Annexure VII** – Parts 2 & 3: Cost Identification at End of Intensive Phase & End of Treatment.
(DOCX)

## Acknowledgments

We would like to thankthe Senapati District Student & Women Organizations (SAWOs) along with the federationunit leaders and their team for not only their valuable comments &suggestions while developing the study protocol to meet the local needs,but also for their continuous help & support in the implementation of the study and their commitment to end TB from the district. We would also like to express our deepest appreciationto the Directorate of Health Service Manipur, NTEP Manipur, the Senapati District Administrators, district health teams, all villages' leaders, and the churches Pastors for their continuous help and support in implementing the study. We are deeply indebted to the project key stakeholders members from ICMR-NIRT, ICMR HQrs., the Central TB Division, WHO India Country Office and the external subject expert members of ICMR for their critical review and evaluation of the study proposal. We extend special thanks to Mr. A Saloni Tony, President of SDSA, for his active participation and for motivating the student federation units to engage tribal communities in the study. We are also thankful to all the study participants', the project staff involved in the study

(Research Scientists, field investigator, Lab Technicians, Data Entry Operators, senior project Assistant), including the Project Officer (Amit Solanki) for generous help in assisting & revising the manuscript and for all budget related processing. We are indebted to all well-wishers for the successful implementation of our project.

## Author contributions

**Conceptualization:** Arangba S, Singh S, Nagarajan K, Malaisamy M, Watson B, Muanching L, Elangbam V, Ngade D, Ngaopuo A, Lungnalii KT, Serto T, Nair D, Vignes Anand S, Elizabeth RK, Mark PS, Hanah RN, Yonuo P, Padma Priyadarshini C, Harpreet Kaur.

**Data curation:** Watson B.

**Funding acquisition:** Arangba S, Harpreet Kaur.

**Investigation:** Arangba S, Nagarajan K, Malaisamy M, Muanching L, Mattoo SK, Elangbam V, Singh WS, Ngade D, Ngaopuo A, Lungnalii KT, Serto T, Pfoze P, Nair D, Elizabeth RK, Mark PS, Hanah RN, Percy S, Harpreet Kaur.

**Methodology:** Arangba S, Singh S, Nagarajan K, Malaisamy M, Watson B, Muanching L, Mattoo SK, Elangbam V, Singh WS, Serto T, Pfoze P, Nair D, Vignes Anand S, Elizabeth RK, Mark PS, Yonuo P, Percy S.

**Project administration:** Arangba S, Singh S, Nagarajan K, Malaisamy M, Watson B, Muanching L, Singh WS, Ngaopuo A, Lungnalii KT, Serto T, Pfoze P, Vignes Anand S, Elizabeth RK, Hanah RN, Yonuo P, Percy S, Padma Priyadarshini C, Harpreet Kaur.

**Resources:** Arangba S, Singh S, Watson B, Muanching L, Singh WS, Padma Priyadarshini C, Harpreet Kaur.

**Supervision:** Arangba S, Singh S, Nagarajan K, Watson B, Muanching L, Mattoo SK, Elangbam V, Ngade D, Ngaopuo A, Lungnalii KT, Serto T, Nair D, Vignes Anand S, Elizabeth RK, Mark PS, Hanah RN, Yonuo P, Percy S, Padma Priyadarshini C, Harpreet Kaur.

**Writing – original draft:** Arangba S, Nagarajan K, Malaisamy M, Watson B.

**Writing – review & editing:** Arangba S, Singh S, Nagarajan K, Malaisamy M, Watson B, Muanching L, Mattoo SK, Elangbam V, Singh WS, Ngade D, Ngaopuo A, Lungnalii KT, Serto T, Pfoze P, Nair D, Vignes Anand S, Elizabeth RK, Mark PS, Hanah RN, Yonuo P, Percy S, Padma Priyadarshini C, Harpreet Kaur.

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
