## [Decision Letter · Decision Letter 0]

Dear Dr. Kaur,

Thank you for submitting your manuscript to PLOS ONE. After careful consideration, we feel that it has merit but does not fully meet PLOS ONE’s publication criteria as it currently stands. Therefore, we invite you to submit a revised version of the manuscript that addresses the points raised during the review process.

We look forward to receiving your revised manuscript.

Kind regards,

Zewdu Gashu Dememew, M.D

Academic Editor

PLOS ONE

Dear Authors,

Congratulation to come up with quite relevant protocol for publication.

Please just attend to the relevant comments and suggestions given from peer reviews. Fortunately, a lot of reviewers were interested in the review of your protocol.

Good luck!

Zewdu

Reviewers' comments:

Reviewer's Responses to Questions

**Comments to the Author**

1. Does the manuscript provide a valid rationale for the proposed study, with clearly identified and justified research questions?

Reviewer #1: Yes

Reviewer #2: Yes

Reviewer #3: Partly

Reviewer #4: Yes

Reviewer #5: Yes

Reviewer #6: Yes

Reviewer #7: Partly

Reviewer #8: Yes

Reviewer #9: Yes

Reviewer #10: Yes

2. Is the protocol technically sound and planned in a manner that will lead to a meaningful outcome and allow testing the stated hypotheses?

Reviewer #1: Yes

Reviewer #2: Yes

Reviewer #3: Partly

Reviewer #4: Yes

Reviewer #5: Yes

Reviewer #6: Yes

Reviewer #7: Partly

Reviewer #8: Yes

Reviewer #9: Partly

Reviewer #10: Yes

3. Is the methodology feasible and described in sufficient detail to allow the work to be replicable?

Reviewer #1: Yes

Reviewer #2: Yes

Reviewer #3: No

Reviewer #4: Yes

Reviewer #5: Yes

Reviewer #6: Yes

Reviewer #7: No

Reviewer #8: Yes

Reviewer #9: Yes

Reviewer #10: No

4. Have the authors described where all data underlying the findings will be made available when the study is complete?

Reviewer #1: Yes

Reviewer #2: Yes

Reviewer #3: No

Reviewer #4: Yes

Reviewer #5: Yes

Reviewer #6: Yes

Reviewer #7: Yes

Reviewer #8: Yes

Reviewer #9: Yes

Reviewer #10: Yes

5. Is the manuscript presented in an intelligible fashion and written in standard English?

Reviewer #1: Yes

Reviewer #2: Yes

Reviewer #3: Yes

Reviewer #4: Yes

Reviewer #5: Yes

Reviewer #6: Yes

Reviewer #7: Yes

Reviewer #8: Yes

Reviewer #9: No

Reviewer #10: Yes

You may also provide optional suggestions and comments to authors that they might find helpful in planning their study.

Reviewer #1: India remains the leading contributor to global burden of tuberculosis and it is obvious that in order to reduce global TB burden we need to reduce TB burden in India. At the same time in order to achieve END-TB goals in India we need to address TB diagnostics and treatment issues, including TB gaps in case finding, in every region and district in India, what will eventually result in reduction of incidence and mortality of TB in whole country.

The manuscript provides lots of information and is a great example of national TB program activities in India.

Few comments:

1. Are volunteers involved in active case finding provided with personal respiratory protection (FFP2/FFP3 respirators) since they might be exposed to lots of presumptive TB patients?

2. Does the protocol involve active contact tracing or it is not a part of the study, but has already been implemented by NTP in India? To clarify, there is and will be information on cases already registered in the district and contacts from those cases will be known to the NTP identifying certain households as focus of TB infection and they should be targeted for LTBI/TB diagnostics. Will that work be done within the proposed protocol?

3. Following previous question – what are the follow-up actions if TB case , especially smear positive highly contagious, is identified throughout the study in the household ? will the be a return visit for LTBI screening and possible LTBI treatment?

4. Will rifampicin resistance/MDR be tested in cases throughout the study? Will treatment for DR TB be provided for those patients?

Reviewer #2: My general comments:

In general, the authors presented a study protocol (a quasi-experimental pre-post study) for active case finding and treatment of tuberculosis (TB), focusing on inhabitants aged >15 years residing in the Senapati District, Manipur, India. This study aims to engage the Student and Women Organizations (SAWOs) in augmenting the TB case finding and treatment, based on clinical symptoms, chest xray and sputum testing (using rapid molecular test-CBNAAT). The rationale for this study is very good.

My first concern is that the active case finding (ACF) and treatment for TB elimination and engaging organizations to enhance the effectiveness of ACF is a novel idea. This has been performed and published by Australian Research Teams (Guy B.Marks, Greg J.Fox et al., https://www.nejm.org/doi/full/10.1056/NEJMoa1902129). My suggestion is that the authors should reconsider the word “novel community-led strategy revolutionizing…”. I think “scaled up” is better term.

My second concern is the significance of the gold standard for testing TB so that the comparison outcomes (TB case notification rate, adherence rate, and treatment outcome) between the intervention and control arms are more accurate. The rapid molecular test CBNAAT has been reported to have a sensitivity of 63% to 75% ((https://pmc.ncbi.nlm.nih.gov/articles/PMC10654686/), indicating that approximately 25% to 37% of TB cases would be missed if only using the rapid molecular test CBNAAT. From my clinical and research knowledge, there will be a certain proportion of TB patients with trace calls by molecular TB testing ( about 50%-60% reveals negative TB culture). I think that there should be included TB culture (MGIT), regarding the true TB population with false negative CBNAAT and/or normal chest xray.

I highly believe that the TB culture (MGIT method…), applied for negative TB molecular testing with clinical and radiological findings consistent with tuberculosis, will ensure that the overall outcome comparisons between intervention and control arms are best accurate.

Reviewer #3: In this study protocol, the authors intend to study the impact of student and women organisations in improving TB case notifications. India has committed to end TB by 2025, this research seems to be late whereas such an intervention should have been a part of the program. Also, the protocol paper comes at a time when the sample collection is nearing completion.

Notwithstanding the above, as far as the protocol is concerned, there are a few issues that the authors need to clarify and address:

Details of the district including the health infrastructure both public and private, community-based organisations working for TB etc need to be elaborated for the reader to have an understanding the necessity of the intervention.

In the data from national program shared in Table 1 on TB notifications, adherence and outcomes it is observed that the program has achieved >100% of the target. It is strange that the authors mention they by this intervention achieve 80% success in the targets. The fact that the program has achieved more than the research proposes makes this study redundant.

The exclusion of the highly vulnerable populations such as mobile population/visitors, institutional populations (schools, colleges, offices, prisons, defence establishments, hospitals, nursing homes, hostels) and areas where survey operations are considered not to be feasible like insecurity areas is not clearly explained as large numbers of TB cases can be missed by such exclusion. The aim is to achieve a “comprehensive” approach to TB elimination.

Additionally, the authors have restricted the term “comprehensive approach” to the cascade of TB care whereas “comprehensive” would entail involvement of ALL stakeholders in the TB cascade such as private health care providers, NGOs/CBOs, TB champions etc.

Baseline assessment need to include anthropometric and nutrition data which are the current focus of all TB programs.

As per recommendation after symptom screening, X-ray screening comes next and then microbiological examination. This especially in a community setting and with the recent emphasis on asymptomatic sub clinical TB.

Linkages with the NTEP need to be elaborated in more detail in line 250.

Since the study is nearing completion, it would be good to provide interim data on achievement so far.

Lines 340, 341 mentioning direct costs, the authors mention doctor’s consultation fees, x-ray and sputum exam expenses - Where does this come in? cases are directly referred to NTEP!

Line 421, there is a mention of barriers to access to health care – was this studied in the study area what were the findings, if not this should be part of the study protocol.

Line 422, authors mention that the study addresses the social and behavioural factors that influence disease transmission as well as treatment outcomes by involving community members as active participants in TB control activities. What were these factors the authors wish to study? Were TB champions also involved in the process?

Line 435, the authors mention about the comprehensive approach. However, what they describe is the cascade of care and not a comprehensive approach which would entail involving all stakeholders in the process of ACF as mentioned above. Authors can refer to the many publications in the Indian context.

Reviewer #4: The objectives of the study protocol are clearly stated and ethically sound. Description of the research stages well described.

Would it be possible to mention the population size (15 y.o and above) of the Senapati district as according to the protocol approximately 316 volunteers will be involved to the activities ( 1 person per 150 households; 47,411 households in total) and later how many people were reached by volunteers, please?

On the implementation of the ACF phase, I would suggest to write more about Infection control measures for volunteers while making the screening, collecting sputum samples and packing specimens for the transportation to the laboratory. How the transportation of the specimen will be organized ( who transport, how, the distance to the nearest CB-NAAT point)

Although the study is ongoing, it is a second round of the ACF, I am looking forward seeing the results of the study!

Reviewer #5: The novelty of the concept may appear limited, as community systems for TB case finding have been previously documented in countries such as Ethiopia, South Africa, and Kenya, which the authors may wish to reference. Additionally, patient support group initiatives in Kerala bear some resemblance to this proposal, though these are more specifically targeted towards diagnosed patients.

A key contribution of this research could lie in its quantification of community systems—a measurement that has not yet been fully developed.

The authors should also consider providing a rationale for the target adjustments made by the National TB Program (NTP) over the years. This will support accurate impact estimation, as target adjustments are often subjectively set. Collaborating with WHO and the NTP team could help establish a reliable baseline, as solely relying on NTP targets might lead to skewed analysis.

Reviewer #6: This study is important highest TB burden country and the study is well designed and recommend to accept for publication.

Reviewer #7: 1. Methodology section is incomplete

2. CRF components to be mentioned in the methodology section

3. Difference between ACF by NTEP and ECF executed by this project to be mentioned in detail. Also the uniqueness and novelty of the study in terms of community engagement and its difference from the existing community outreach programs by NMC to be mentioned in detail

4. Target population to be defined precisely

5. Any similar studies done in other countries or other programs may be quoted as reference

Reviewer #8: Major Comments

Page 3, Line 33-36: The introduction needs better contextualization of the TB burden in Manipur. Specific local challenges could be highlighted in more detail, beyond the general statement about India's TB burden. This would strengthen the rationale for the study.

Page 4, Line 77-84: The study timeline is mentioned to be extended by 6 months due to disruptions, but the impact of these disruptions on study outcomes is not fully discussed. Please elaborate on how these events might affect the findings, participant retention, or data collection.

Page 5, Line 107-110: The TB burden in Senapati district is noted as unknown, yet a similar neighboring district's TB burden is cited. Consider providing more data or clarifying why exact prevalence estimates are unavailable. This would support the choice of study site and methodology.

Page 15, Line 373-374: The methodology lacks detail on how potential confounders will be identified and controlled in the logistic regression analysis. Please specify the strategies for confounder identification and handling, as this is crucial for the robustness of the results.

Minor Comments

Page 2: A more detailed definition of the role of Student and Women Organizations (SAWOs) would benefit readers unfamiliar with these groups.

Page 7, Line 158: Typo in 'volunteers give written consent' - 'give' should be 'gave'. Please correct verb tense consistency.

Reviewer #9: The protocol does indeed present a novel approach to TB case finding and overall TB control in Senapati District of Manipur,India.

The use of student and women local organisations offers an opportunity for ownership of the intervention by the community and some assurance of sustainability after project life

The protocol is presented soundly and manages to put across activities that will carried out during project life.

However,the authors could consider the following to make the protocol more clearer and avoid confusion with the readers;

For example,line 196-198 talks about village mapping and assigning of unique identification numbers to participants according to the designated area. It will be helpful for the authors to describe and show how this process will be implemented

Line 215-217 talks about screening tools…the authors should state that a symptom screening tool will be used other and state any other tool will be used if any

Line 393-399,the authors describe payments/stipends that will be given to the volunteers.

• An honorarium of ₹500 per month

• mobile recharge of ₹700 per quarter

• NTEP schemes NTEP ;

informant incentive (₹500/-

Patient support incentive;₹1000 for drug sensitive TB patients and ₹5000 for MDR-TB

It would helpful for the authors to provide context and justification for these payments and how they circle back to sustainability after project life.

Overally,this protocol could benefit from a copy editor to make readability and flow of the protocol better.

Reviewer #10: The manuscript describes an interesting approach to TB case finding in a vulnerable and a violence prone area in East India. The effort to undertake it is laudable. This reviewer offers the following comments.

1. At the onset it would be desirable to know why 64% of TB symptomatics do not seek treatment. The Project should aim to elicit some responses from the TB cases detected in the study to guide mitigation strategies in the location context.

2. The method section needs to elaborate on the approach to handling of asymptomatics between the 2 phases of intervention

3. It is unclear as to who has designed the costing tool and how is the costing data going to be analyzed. The researcher team needs to be well trained in eliciting responses to costing questions which would be best undertaken in the paper format.

4. The methods section also needs to describe the deployment of human resources for each household and estimate the time taken to complete the activities for identification of TB cases at the household level. Would be useful for future studies using a similar approach.

5. It should be clarified as to what were the selection criteria if any for the SAWO volunteers

6. The interim time period between the 2 intervention phases of 6 months appears on the lower side. A period of 12 months may have been more productive. In between the SAWO volunteers could be engaged in community health education activities to supplement case finding.

7. Fig.1 is indicative of good outcome indices. Would the present initiative be able to better these outcomes significantly. If any other foreseen tangible benefits to the community are identified, it would be helpful to mention them.

8. Reasons for refusal to participate in the survey should also be recorded and analysed.

We look forward to the results arising from this approach which maybe further strengthened by observations on its sustainability.

9. In the section on exposure to risk factors (line 118), the level of undernutrition in the community needs mentioning in view of the low SES and the high prevalence of TB.

10.Taking as an example the questionnaire on TB KAP Section 2 and 3 words like "drug resistant TB", virus, bacteria, fungi (Section 2) and "public health problem" (section 3) should be replaced in a way that is understood by the local population.

**Do you want your identity to be public for this peer review?** For information about this choice, including consent withdrawal, please see our Privacy Policy

Reviewer #1: No

Reviewer #2: **Yes: ** Thanh Tat Nguyen

Reviewer #3: No

Reviewer #4: No

Reviewer #5: **Yes: ** Shibu Vijayan

Reviewer #6: No

Reviewer #7: **Yes: ** Dr.Krithikaa Sekar

Reviewer #8: No

Reviewer #9: No

Reviewer #10: No

---

## [Author Response · Author response to Decision Letter 1]

4 Apr 2025

I. Peer Review for MEDIN Manuscript PONE-D-24-38779

Date: 22-October-2024

“Breaking barriers for TB elimination: A novel community-led strategy revolutionizing Tuberculosis case finding and treatment support in Senapati District of Manipur-A quasi-experimental pre-post study protocol”

These are my comments in the peer review for the manuscript requested.

My general comments:

In general, the authors presented a study protocol (a quasi-experimental pre-post study) for active case finding and treatment of tuberculosis (TB), focusing on inhabitants aged >15 years residing in the Senapati District, Manipur, India. This study aims to engage the Student and Women Organizations (SAWOs) in augmenting the TB case finding and treatment, based on clinical symptoms, chest xray and sputum testing (using rapid molecular test-CBNAAT). The rationale for this study is very good.

Commnet 1. My first concern is that the active case finding (ACF) and treatment for TB elimination and engaging organizations to enhance the effectiveness of ACF is a novel idea. This has been performed and published by Australian Research Teams (Guy B.Marks, Greg J.Fox et al., https://www.nejm.org/doi/full/10.1056/NEJMoa1902129). My suggestion is that the authors should reconsider the word “novel community-led strategy revolutionizing…”. I think “scaled up” is better term.

Response 1. We had used the term novel in the title, based on the existing literature, as we are involving the organized Student And Women Organizations (SAWOs) in TB case finding or in providing support during treatment. Though previous studies have engaged community members such as Community Healthcare Workers (CHWs), Accredited Social Health Activists (ASHAs), National Service Scheme (NSS) students and elementary and secondary school students, these individuals did not operate within a formal organizational structure that contributes to village-level decision-making on broader issues.

The SAWOs engaged in this project are well-structured organizations with clearly defined roles and responsibilities. They represent youth & women from their respective villages, communities, districts and collectively participate in larger forums to advance common goals and objectives. Leadership positions within these organizations are typically given to active and influential members of the group, chosen through an election process.

Each village, church and community (both tribal and non-tribal) has its own youth and women organizations. Members of student organizations include those who are currently pursuing their studies, or have either completed their education or have dropped out. However, most student leaders are individuals who have completed their education and are not currently employed.

Similarly, members of women organizations are primarily married women including widows, divorcees as well as those living with their husband and children. This project is built on the four pillars of India's National Strategic plan that is Detect, treat, and prevent, Build. This project tries to address the cascade of care from educating people about TB to the identification of TB cases to the completion of the treatment. This study brings a novel and comprehensive approach to addressing TB in the community by integrating SAWO engagement, cost-effectiveness analysis and a participatory research design. These elements set it apart from both NTEP’s ACF, which is more focused on diagnostic surveillance. By fostering community ownership, addressing all pillars of the END TB strategy and evaluating the economic impact, our study offers a holistic, inclusive and evaluative model for improving TB controls that could provide valuable insights for future TB programs.

In view of the foregoing rationale of engaging a unique & distinctly placed active and influential community group, operating from within a formal organizational structure, who are elected through a democratic process and contribute to village-level decision making on broader issues, and thus offer an opportunity for ownership of the intervention and some assurance of sustainability after project life, the authors think the study to be novel and thus request the Reviewer if novelty can be retained in the title of the manuscript, considering this group as a novel one, which is not the same as used in the referred publication by Australia research team.

Comment 2. My second concern is the significance of the gold standard for testing TB so that the comparison outcomes (TB case notification rate, adherence rate, and treatment outcome) between the intervention and control arms are more accurate. The rapid molecular test CBNAAT has been reported to have a sensitivity of 63% to 75% ((https://pmc.ncbi.nlm.nih.gov/articles/PMC10654686/), indicating that approximately 25% to 37% of TB cases would be missed if only using the rapid molecular test CBNAAT. From my clinical and research knowledge, there will be a certain proportion of TB patients with trace calls by molecular TB testing ( about 50%-60% reveals negative TB culture). I think that there should be included TB culture (MGIT), regarding the true TB population with false negative CBNAAT and/or normal chest xray.

Response 2. We agree with the Reviewer’s concern. However, the study district is a hard-to-reach area, and the testing is adopted as per recommendations of NTEP to use the molecular method as the first diagnostic tool to diagnose TB in this area, so that we can detect TB and DR-TB without delay. All our interventions have been implemented as per the routine NTEP activities except involving the volunteers from each village to carry out the activities. As per the local TB program, CBNAAT has been used as the first diagnostic tool to diagnose the disease followed by chest X-Ray for those individuals whose sputum results turn out to be negative. Similarly, we are doing it in a programmatic mode to maintain the accuracy of our intervention (pre-and post). The suggestion is 100% valid that we may miss approximately 25% to 37% of TB cases. However, adding culture (MGIT) will become a new intervention but the local NTEP is not doing it, and this may be considered as a limitation of the study, as this will further delay the study completion, which is both time bound and financially constrained. We will however recommend this to NETP to add culture / MGIT and make it program inclusive.

Comment 3. I highly believe that the TB culture (MGIT method…), applied for negative TB molecular testing with clinical and radiological findings consistent with tuberculosis, will ensure that the overall outcome comparisons between intervention and control arms are best accurate.

Response 3. Though we are not doing culture for those CBNAAT-negative samples as per the local TB program. However, we are referring those individuals whose sputum turns out to be negative in the CBNAAT test to the nearest chest X-Ray centre for further investigations, as mentioned in the revised manuscript, line numbers 289 to 295.

My specific comments:

1. Title:

The title needs to be reconsidered, regarding the term “novel community-led strategy revolutionizing….”, because active case finding TB and treatment and engaging organizations is not a breakthrough.

Response: We had used the term novel in the title, based on the existing literature, as we are involving the organized Student And Women Organizations (SAWOs) in TB case finding or in providing support during treatment. Though previous studies have engaged community members such as Community Healthcare Workers (CHWs), Accredited Social Health Activists (ASHAs), National Service Scheme (NSS) students and elementary and secondary school students, these individuals did not operate within a formal organizational structure that contributes to village-level decision-making on broader issues.

The SAWOs engaged in this project are well-structured organizations with clearly defined roles and responsibilities. They represent youth & women from their respective villages, communities, districts and collectively participate in larger forums to advance common goals and objectives. Leadership positions within these organizations are typically given to active and influential members of the group, chosen through an election process. Each village, church and community (both tribal and non-tribal) has its own youth and women organizations. Members of student organizations include those who are currently pursuing their studies, or have either completed their education or have dropped out. However, most student leaders are individuals who have completed their education and are not currently employed. Similarly, members of women organizations are primarily married women including widows, divorcees as well as those living with their husband and children.

This project is built on the four pillars of India's National Strategic plan that is Detect, treat, and prevent, Build. This project tries to address the cascade of care from educating people about TB to the identification of TB cases to the completion of the treatment.

This study brings a novel and comprehensive approach to addressing TB in the community by integrating SAWO engagement, cost-effectiveness analysis and a participatory research design. These elements set it apart from both NTEP’s ACF, which is more focused on diagnostic surveillance. By fostering community ownership, addressing all pillars of the END TB strategy and evaluating the economic impact, our study offers a holistic, inclusive and evaluative model for improving TB controls that could provide valuable insights for future TB programs.

In view of the foregoing rationale of engaging a unique & distinctly placed active and influential community group, operating from within a formal organizational structure, who are elected through a democratic process and contribute to village-level decision making on broader issues, and thus offer an opportunity for ownership of the intervention and some assurance of sustainability after project life, the authors think the study to be novel and thus request the Reviewer if novelty can be retained in the title of the manuscript, considering this group as a novel one, which is not the same as used in the referred publication by Australia research team.

2. Abstract: The abstract is clear, succinct, and well-written,

3. Introduction:

The introduction is very well written and clearly describes the TB burden in India and the rationale for this study to be conducted. The aims of this study have clearly been stated.

4. Materials and Methods:

The methods are well designed, structured and written.

4.1 I suggest that the authors think more about the number of TB patients with false negative rapid molecular test-CBNAAT (indeed, they have true TB infection) and trace calls from molecular testing (all estimated to account for 20%-30%) of all people tested with molecular testing. Further tuberculosis cultures (for example, with MGIT method) should be performed. This will ensure that the outcome comparisons between the intervention and control arms are accurate

Response: We agree with the Reviewer’s concern. However, the study district is a hard-to-reach area, and the testing is adopted as per recommendations of NTEP to use the molecular method as the first diagnostic tool to diagnose TB in this area, so that we can detect TB and DR-TB without delay. All our interventions have been implemented as per the routine NTEP activities except involving the volunteers from each village to carry out the activities. As per the local TB program, CBNAAT has been used as the first diagnostic tool to diagnose the disease followed by chest X-Ray for those individuals whose sputum results turn out to be negative. Similarly, we are doing it in a programmatic mode to maintain the accuracy of our intervention (pre-and post). The suggestion is 100% valid that we may miss approximately 25% to 37% of TB cases. However, adding culture (MGIT) will become a new intervention but the local NTEP is not doing it, and this may be considered as limitation of the study, as this will further delay the study completion, which is both time bound and financially constraint. We will however recommend this to NETP to add culture / MGIT and make it program inclusive.under ‘My General Comments’.

4.2 I also think the authors should clarify the sampling method in village mapping and household enumeration (Lines 186 and 193). It should be randomly selected using a computerized generated sampling method in predetermining villages for intervention and standard-care arms. These factors are significant for preventing potential selection bias.

Response: Our study is a pre-post study design, which involves screening of all the villages of the study district (as highlighted in line numbers 206 -213) village mapping is being carried out for the whole villages of the district, and household enumeration is done based on the study eligibility criteria for the whole district population. Hence there is no selection bias in the sampling method in village mapping and household enumeration.

4.3 Regarding the study population, the exclusion of mobile population/visitors and institution populations (Lines 135-138) should be carefully considered. On the one hand, this will enhance survey completion and retention rates. On the other hand, these populations can further transmit the tuberculosis (in cases with undiagnosed active TB), moving from this village to other villages of the study sites. As a result, this will influence the primary study outcomes of TB notification rates and outcomes at a certain level.

From my personal view, I recommend including mobile populations, regarding my above consideration in balance with maximizing the retention rate in the study, generalizability, and optimizing the number of population receiving TB treatment and reducing cross-transmission. However, the authors should consider this point further.

Response: Since our project covers all the villages of Senapati district, intra-district migration from one village to another in the same district (i.e., within district migration) issues have been taken care and we are able to include such mobile populations. Further, based on our enumeration and 1st round of screening experiences even after excluding the mobile population, we came across a huge population difference in the enumerated population and the number of people who have been screened during the 1st round of screening. We have randomly selected a few villages to cross verify the reasons for not being able to be screened in the 1st round of screening after being included during the enumeration. We found that a lot of people migrate to other districts and states for work/studies. However, we cannot capture such mobile populations, which move from the study district to another district/state and this becomes limitation of the study as this is beyond the scope of our study. On the other hand, we could consider including the institution populations like schools, colleges and offices as screening is done early morning and evening. We have made the changes accordingly in our study inclusion criteria in the revised manuscript, line numbers 151-155.

5. Discussion: This is well supported and well written.

My final decision: I think this is a great study and I support this study to be proceeded.

II. Reviewer comments (6)

This study is very important and the study is well designed. I have only minor comments below.

1. It is very important to instruct patients about how to collect quality sputum for the quality diagnosis but author did not mention it.

2. Out of two sputum collected (spot and early morning), which one is used for CBNAAT test?

Comment 1. It is very important to instruct patients about how to collect quality sputum for the quality diagnosis but author did not mention it.

Response 1. We thank the Reviewer for acknowledging our research work.

Regarding the quality sputum collections, we have provided adequate instructions and trained the volunteers during the two-day hands-on training sessions organized on how to extract quality sputum and take care of the several aspects of collecting the sputum samples, which are as follows:

1. To go to open space/well-ve

---

## [Decision Letter · Decision Letter 1]

Breaking barriers for TB elimination: A novel community-led strategy revolutionizing Tuberculosis case finding and treatment support in Senapati District Manipur-A quasi-experimental pre-post study protocol

PONE-D-24-38779R1

Dear Dr. Harpreet Kaur and team,

We’re pleased to inform you that your manuscript has been judged scientifically suitable for publication and will be formally accepted for publication once it meets all outstanding technical requirements.

Kind regards,

Zewdu Gashu Dememew, M.D, PhD

Academic Editor

PLOS ONE

Additional Editor Comments (optional):

Reviewers' comments:

Reviewer's Responses to Questions

**Comments to the Author**

1. Does the manuscript provide a valid rationale for the proposed study, with clearly identified and justified research questions?

Reviewer #3: Yes

Reviewer #4: Yes

Reviewer #7: Partly

Reviewer #9: Yes

Reviewer #10: Yes

2. Is the protocol technically sound and planned in a manner that will lead to a meaningful outcome and allow testing the stated hypotheses?

Reviewer #3: Yes

Reviewer #4: Yes

Reviewer #7: Yes

Reviewer #9: Yes

Reviewer #10: Yes

3. Is the methodology feasible and described in sufficient detail to allow the work to be replicable?

Reviewer #3: Yes

Reviewer #4: Yes

Reviewer #7: Yes

Reviewer #9: Yes

Reviewer #10: Yes

4. Have the authors described where all data underlying the findings will be made available when the study is complete?

Reviewer #3: No

Reviewer #4: Yes

Reviewer #7: Yes

Reviewer #9: Yes

Reviewer #10: Yes

5. Is the manuscript presented in an intelligible fashion and written in standard English?

Reviewer #3: Yes

Reviewer #4: Yes

Reviewer #7: Yes

Reviewer #9: Yes

Reviewer #10: Yes

You may also provide optional suggestions and comments to authors that they might find helpful in planning their study.

Reviewer #3: The authors have addressed all the comments satisfactorily within the constraints of geography, timing, resources and other limitations as indicated in their responses.

Reviewer #4: All questions of reviewers has been answered in an understandable manner. Looking forward on the study results! Good luck!

Reviewer #7: This is a well written manuscript. However, I would like to suggest a few revisions to the manuscript

1. As you had mentioned, SAWOs are already engaged in healthcare community engagement activities and in various welfare programs provided by the government. Hence, this idea may not be entirely "novel" as this is an extension of NTEP activities. So the word "novel" in the title may be reconsidered and alternate terminologies like "repurposed" or "redesignated" may be used.

2. Indicators for monitoring and evaluation of the study may be mentioned methodology section

3. Sample size or Expected prevalence does not compensate for the missed out vulnerable population, who are also the residents of the district. Hence the same may be mentioned.

Thanks and Regards

Dr. Krithikaa Sekar

Reviewer #9: The authors have responded to all the highlighted concerns sufficiently.I have no further concerns with the protocol.

Reviewer #10: The authors have adequately responded to the questions posed by this Reviewer. Some details in Methods could have been added in this protocol paper to guide future studies but hopefully the authors will do so in subsequent publications of the results

**Do you want your identity to be public for this peer review?** For information about this choice, including consent withdrawal, please see our Privacy Policy

Reviewer #3: **Yes: ** Dr. Yatin Dholakia

Reviewer #4: **Yes: ** MD Aiymgul Duishekeeva, Kyrgyzstan

Reviewer #7: **Yes: ** Krithikaa Sekar

Reviewer #9: No

Reviewer #10: No

---

## [Editor Report · Acceptance letter]

PONE-D-24-38779R1

PLOS ONE

Dear Dr. Kaur,

I'm pleased to inform you that your manuscript has been deemed suitable for publication in PLOS ONE. Congratulations! Your manuscript is now being handed over to our production team.

Kind regards,

on behalf of

Dr. Zewdu Gashu Dememew

Academic Editor

PLOS ONE